# Efficient and Accurate Likelihood Estimation via Learning Amortized Adaptive Proposal Distributions

## Abstract

Recent advancements in probabilistic modeling have driven significant progress in deep learning, particularly through the development of generative models based on variational inference. These models optimize a tractable lower bound of the log-likelihood, rather than the log-likelihood itself. However, they often encounter trade-offs between approximation accuracy and computational efficiency. To address these limitations, we propose a novel generative model grounded in importance sampling. Central to our approach is the Amortized Adaptive Proposal Distribution (AAPD), which simultaneously serves as both the proposal distribution for importance sampling and an approximation to the posterior. Extensive evaluations on both synthetic and real-world datasets demonstrate the superior performance and versatility of our method in latent variable modeling. Additionally, we extend our model to mixed-effects settings, effectively addressing some limitations of traditional statistical approaches.

## 1    Introduction

Deep learning has achieved remarkable success in fields such as computer vision (Redmon et al., 2016), natural language processing (Devlin et al., 2019), and speech recognition (Hinton et al., 2012). While the advancements in large-scale data processing and high-performance hardware have contributed to this success, the development of efficient algorithms has also played a crucial role. For instance, architectures such as convolutional neural networks (Lecun et al., 1998), recurrent neural networks (Rumelhart et al., 1988), and transformers (Vaswani et al., 2017) have been pivotal in image analysis (Dosovitskiy et al., 2021), sequential data processing (Dong et al., 2018), and natural language processing (Reimers & Gurevych, 2019). As a result, deep learning is widely recognized as a powerful tool for solving complex problems due to its ability to learn from data and extract important features automatically.

Among deep learning methods, generative models have also advanced rapidly. Their goal is to learn the underlying data distribution and produce realistic samples that reflect its structure. A prominent example is Variational Autoencoder (VAE) (Kingma & Welling, 2014), a probabilistic model characterized by an encoder-decoder structure. A key feature of the VAE is the use of the Evidence Lower Bound (ELBO) for the log-likelihood by introducing a variational distribution to approximate the true posterior distribution. Additionally, Kingma & Welling (2014) proposed the reparameterization trick to improve backpropagation efficiency in the encoder. However, this approach is primarily applicable to Gaussian distributions. For the VAE to fully maximize the log-likelihood, the variational distribution must match the true posterior distribution exactly. Because the family of variational distributions is predefined, such a perfect match is rarely achieved. Consequently, this calls into question whether the VAE genuinely maximizes the log-likelihood.

To overcome this limitation, Qiu & Wang (2021) proposed Adaptive Latent Modeling and Optimization via Neural Networks and Langevin Diffusion (ALMOND), which employs an estimated posterior distribution as the variational distribution. When the model parameters are learned effectively, the variational distribution can always match the true posterior distribution, mitigating some of the limitations described above. While VAE enables efficient posterior sampling through the reparameterization trick, ALMOND relies on the

Unadjusted Langevin Algorithm (ULA) (Roberts & Tweedie, 1996), a highly inefficient sampling method in practice. This stark contrast underscores an inherent trade-off between approximation accuracy and computational efficiency.

To handle these issues, we propose a novel generative model that employs Monte Carlo sampling via importance sampling with amortized adaptive prior learning. The advantages of using importance sampling over ULA and the key contributions of our proposed model are summarized as follows:

- We utilize an importance sampling estimator instead of a mean ergodic estimator (Mattingly et al., 2010) derived from ULA samples. Since it is not ULA-based, initial points and hyperparameters required for ULA are no longer necessary. Furthermore, the importance sampling estimator is unbiased (or has small bias), leading to more accurate estimates compared to ULA.

- The most critical aspect of importance sampling is to choose an appropriate proposal distribution. We propose the Adaptive Amortized Proposal Distribution (AAPD), which integrates variational inference with importance sampling. This approach allows our model to achieve more precise likelihood estimation while enhancing the generation process.

- We illustrate how to apply a generative model across various domains. As a generative model, our model proves effective not only in image generation but also in dimensionality reduction by exploiting latent variables. Moreover, its accurate likelihood estimation stresses a potential for modeling mixed-effect models.

The remainder of the paper is organized as follows: Section 2 reviews related work. Section 3 presents an overview of foundational models, such as VAE and ALMOND, which our proposed model seeks to enhance, followed by the formulation and discussion of our method. In Section 4, we evaluate the proposed model alongside baselines on both synthetic and real-world datasets, providing qualitative and quantitative assessments. Finally, Section 5 concludes with a summary and a discussion of potential future directions.

## 2 Review of literature

**Variational Inference.** Kingma & Welling (2014) laid the foundation for integrating variational inference with deep learning. To better approximate the true posterior distribution with a variational distribution, two primary approaches have emerged. The first is to enhance the expressiveness of the inference network (Rezende & Mohamed (2015); Ho et al. (2020); Lipman et al. (2023)), which lies outside the scope of this study. The second is to develop a more accurate Monte Carlo estimator for the ELBO. A seminal work in this direction is Importance Weighted Autoencoder (IWAE) (Burda et al., 2015), which introduces importance weights into the true log-likelihood, and results in a tighter ELBO. Building on this foundation, subsequent works have proposed tighter ELBOs using sequential Monte Carlo methods (Maddison et al. (2017); Le et al. (2018); Naesseth et al. (2018)) and importance sampling techniques (Thin et al. (2021); Demange-Chryst et al. (2024)). Qiu & Wang (2021) further proposed a novel strategy for tightening the ELBO by parameterizing the variational distribution as an alternative representation of the posterior distribution.

**Applications of generative models.** The power of generative models is demonstrated across various domains, including image generation (van den Oord et al., 2017), natural language generation (Raffel et al., 2020), and audio generation (van den Oord et al., 2016). In particular, latent variables in generative models are useful for downstream machine learning tasks such as clustering (Mukherjee et al., 2019), classification (Li et al., 2019), and dimensionality reduction (Kim & Chun, 2023). Generative models have also been successfully applied to diverse fields, including biology (Wu et al., 2021), image processing (Wang et al., 2024), and finance (Takahashi et al., 2019).

These developments have also influenced statistical methodology. For example, Müller et al. (2019) tackled the classical statistical problem of density estimation by applying normalizing flows. Song et al. (2017) introduced an extraordinary generative model for modeling probability kernels, serving as a generative model-based MCMC methodology. Zhao et al. (2019) attempted to apply VAE to regression in the context of brain aging analysis.

To the best of our knowledge, no study has integrated generative models with Generalized Linear-mixed Models (GLMM). A related study (M.-N. Tran & Kohn, 2020) was the first to apply deep learning to GLMM by substituting (generalized) linear regressors with a neural network. However, it is a simple model with a lack of interpretability and limited discussion on random effects. Simchoni & Rosset (2024) sought to improve the interpretability of random effects by restricting neural network applications to the given covariates while retaining fixed and random effects. Nevertheless, their study primarily focuses on prediction, lacking statistical discussions of these effects.

## 3 Methodology

In this section, we provide a formal review of fundamental models based on variational inference, including the Variational Autoencoder (VAE) and Adaptive Latent Modeling and Optimization via Neural Networks and Langevin Diffusion (ALMOND). We then introduce Amortized Adaptive Importance Sampling (AAIS), which addresses their limitations while leveraging their strengths.

### 3.1 Variational Autoencoder (VAE)

Suppose data $\{x_i\}_{i=1}^N$ is given and the observations are samples from an unknown true distribution $p^*(x)$. We aim to approximate this distribution with a parametrized model $p_\theta(x)$, that is,

$$x_i \sim p_\theta(x) \approx p^*(x)$$

Assuming the existence of a latent variable $z \in \mathbb{R}^d$, the marginal distribution over the observed variables is

$$p_\theta(x) = \int p_\theta(x, z) dz = \int p_\theta(z) p_\theta(x|z) dz,$$

where both $p_\theta(z)$ and $p_\theta(x|z)$ need to be modeled. The optimal parameter $\theta^*$ is obtained by maximizing $p_\theta(x)$. Unfortunately, it cannot be calculated efficiently due to the intractability of $p_\theta(z|x)$. To overcome this we introduce a variational distribution $q_\phi(z|x)$ which is tractable and seeks to approximate $p_\theta(z|x)$, i.e., $q_\phi(z|x) \approx p_\theta(z|x)$. For any choice of the variational distribution $q_\phi(z|x)$, the log-likelihood of $p_\theta(x)$ can be expressed as follows:

$$\log p_\theta(x) = \mathbb{E}_{q_\phi(z|x)}[\log p(x)] = \mathbb{E}_{q_\phi(z|x)}[\log \frac{p_\theta(x, z)}{p_\theta(z|x)}]$$

$$= \mathbb{E}_{q_\phi(z|x)}[\log \frac{p_\theta(x, z)}{q_\phi(z|x)}] + \mathbb{E}_{q_\phi(z|x)}[\log \frac{q_\phi(z|x)}{p_\theta(z|x)}] \tag{1}$$

We denote each term in Equation (1) by $\mathcal{L}(\theta, \phi; x)$ and $D_{KL}(q_\phi(z|x)||p_\theta(z|x))$, respectively. Then,

$$\log p_\theta(x) = \mathcal{L}(\theta, \phi; x) + D_{KL}(q_\phi(z|x)||p_\theta(z|x)) \tag{2}$$

The second term in Equation (2) is called the Kullback-Leibler (KL) divergence of $q_\phi(z|x)$ from $p_\theta(z|x)$, which is always non-negative. As a result,

$$\log p_\theta(x) \geq \mathcal{L}(\theta, \phi; x) = \mathbb{E}_{q_\phi(z|x)}[\log p_\theta(x|z)] - D_{KL}(q_\phi(z|x)||p_\theta(z)), \tag{3}$$

where $\mathcal{L}(\theta, \phi; x)$ is called the Evidence Lower Bound (ELBO). By maximizing it, we expect $\log p_\theta(x)$ to be maximized as well. The derivations above are referred to as variational inference. It is worth noting that the networks $q_\phi(z|x)$ and $p_\theta(x|z)$ resemble data compression/generation. Since this kind of model has been called an autoencoder, we call the proposed model variational autoencoder (VAE). More information on VAE can be found in the original paper by Kingma & Welling (2014).

For optimization, Kingma & Welling (2014) used Monte Carlo sampling to estimate the expectation in Equation (3)

$$\mathbb{E}_{q_\phi(z|x)}[\log p_\theta(x|z)] \approx \frac{1}{M} \sum_{i=1}^M \log p_\theta(x|z_i), \tag{4}$$

where $\{z_i\}_{i=1}^M$ are i.i.d. samples from $q_\phi(z|x)$. Since the KL divergence between two normal distributions has a closed form, they suggested that $q_\phi(z|x)$ belongs to the family of Gaussian distributions with diagonal covariance matrices and assumed that $p_\theta(z)$ is the standard multivariate Gaussian distribution on $\mathbb{R}^d$. Then, the KL divergence in Equation (3) can be simplified as:

$$D_{KL}(q_\phi(z|x)||p_\theta(z)) = -\frac{1}{2}\sum_{i=1}^d[1 + \log\sigma_\phi^2(x)_i - \mu_\phi(x)_i^2 - \sigma_\phi^2(x)_i], \tag{5}$$

if $q_\phi(z|x) \sim \mathcal{N}(\mu_\phi(x), \mathrm{Diag}(\sigma_\phi^2(x)_1, ..., \sigma_\phi^2(x)_d))$ and $p_\theta(z) \sim \mathcal{N}(0, I_d)$

where $\mu_\phi(x)_i$ denotes the $i$-th component of the mean vector $\mu_\phi(x)$, and $\sigma_\phi^2(x)_i$ refers to the $i$-th diagonal element of the diagonal covariance matrix $\mathrm{Diag}(\sigma_\phi^2(x)_1, ..., \sigma_\phi^2(x)_d)$. The subscript $\phi$ indicates that both the mean vector and the covariance matrix are parameterized by the neural network parameters $\phi$.

While these assumptions improve computational efficiency, they may hinder log-likelihood maximization. For a fixed $\theta$, an increase in the ELBO in Equation (3) is equivalent to a decrease in the KL divergence in Equation (2). An equality in Equation (3) holds if and only if $q_\phi(z|x) = p_\theta(z|x)$, which does not occur unless the true posterior distribution belongs to the family of Gaussian distributions with diagonal covariance matrices. Therefore, it remains questionable whether VAE can truly maximize the log-likelihood.

## 3.2 Adaptive Latent Modeling and Optimization via Neural Networks and Langevin Diffusion (ALMOND)

Qiu & Wang (2021) pointed out the restriction of VAE mentioned in Section 3.1 and introduced another type of variational distribution to make the lower bound tighter

$$q_\phi(z|x) := p_\eta(z|x),$$

which means that the posterior distribution parameterized by $\eta$ is regarded as a variational distribution. What are the potential benefits? First, the ELBO corresponding to the variational distribution becomes

$$\log p_\theta(x) \geq \mathcal{L}(\theta, \eta; x) \tag{6}$$
$$= \mathbb{E}_{p_\eta(z|x)}[\log p_\theta(x|z)] - D_{KL}(p_\eta(z|x)||p_\theta(z)), \tag{7}$$

where $p_\theta(z) \sim \mathcal{N}(0, I_d)$. An important property of (6) is that $\log p_\theta(x) = \mathcal{L}(\theta, \eta; x)$ holds if $p_\eta(z|x) = p_\theta(z|x)$ (that is, $\eta = \theta$). As long as the model is trained correctly, the ELBO is maximized and simultaneously touches the equality in Equation (6). That is, for a fixed $\eta$,

$$\mathcal{L}(\theta, \eta; x) \leq \log p_\theta(x) \quad \text{for all } \theta, \tag{8}$$
$$\mathcal{L}(\eta, \eta; x) = \log p_\eta(x) \tag{9}$$

This property follows from the choice of variational distribution and introduces a subtle distinction from VAE. While VAE trains the variational distribution $q_\phi(z|x)$ to approximate the posterior $p_\theta(z|x)$, the proposed model estimates the posterior distribution simply by exploring the same parameter space.

---

**Algorithm 1** Unadjusted Langevin Algorithm (ULA)

---

1: Initialize a point $z_0$, step size $s$, and number of iterations $L$
2: **for** $i = 1$ to $L$ **do**
3:     Sample $\epsilon_i$ from the standard normal distribution
4:     Update:

$$z_i \leftarrow z_{i-1} + s\frac{\partial}{\partial z}\log p_\eta(z \mid x) + \sqrt{2s}\epsilon_i$$

5: **end for**
6: **return** $z_L$ as a sample drawn from $p_\eta(z \mid x)$

---

In the aspect of optimization, since samples from $p_\eta(z|x)$ are not easily accessible, the estimation (4) becomes infeasible. In addition, the KL divergence in Equation (7) no longer has a closed form, even when $p_\theta(z)$ is a normal distribution. However, the properties (8) and (9) satisfy the requirements for the minorize-maximization (MM) algorithm (Sun et al., 2017). Consequently, we do not need to optimize $\mathcal{L}(\theta, \eta; x)$ with respect to $\eta$, and the remaining concern is how to deal with the expectation in Equation (7). To this end, Qiu & Wang (2021) adopted the mean ergodic method (Mattingly et al., 2010). Formally,

$$\mathbb{E}_{p_\eta(z|x)}[\log p_\theta(x|z)] \approx \frac{1}{M} \sum_{i=1}^{M} \log p_\theta(x|z_i), \tag{10}$$

where $\{z_i\}_{i=1}^{M}$ are samples from $p_\eta(z|x)$ generated by Unadjusted Langevin Algorithm (ULA; see Algorithm 1). The proposed model is named Adaptive Latent Modeling and Optimization via Neural Networks and Langevin Diffusion (ALMOND). Note that the samples $z_i$ are not independent, and thus the estimator (10) is biased compared to the one in Equation (4). For theoretical analysis of this estimator, refer to the original work (Qiu & Wang, 2021).

---

**Algorithm 2** ALMOND with the warm-start method

---

1: Initialize $\theta_0$, Langevin samples $\{z_i^{(0)}\}_{i=1}^{M}$ for $x$, and the associated hyperparameters for Algorithm 1
2: **while** $\mathcal{L}(\theta_n, \theta_n; x)$ is not converged **do**
3:     $g(\theta) \leftarrow \mathcal{L}(\theta, \theta_n; x)$
4:     **for** $i = 1$ to $M$ **do**
5:         $z_i^{(n+1)} \leftarrow \text{ULA}(z_i^{(n)})$                                             ▷ See Algorithm 1
6:     **end for**
7:     $g(\theta)$ is estimated by the ergodic estimator (10) on $z_1^{(n+1)}, \ldots, z_M^{(n+1)}$
8:     $\theta_{n+1} \leftarrow \arg\max_\theta g(\theta)$
9: **end while**
10: **return** an optimal parameter $\theta^*$

---

Despite its effectiveness, ULA has several drawbacks. First, it requires multiple hyperparameters to be used in the ALMOND implementation, and the resulting samples are highly sensitive to these values (Roberts & Tweedie, 1996). Second, each transition to a new point depends on the gradient of $p_\eta(z|x)$, which may lead to incorrect transitions if the decoder is not adequately trained. To remedy this, Qiu & Wang (2021) suggested pre-training a VAE beforehand and then using it as the initial model of ALMOND. This procedure has the shortcoming of becoming a two-stage model. Lastly, enhancing the convergence speed and sample quality of ULA remains a critical challenge. They proposed the warm-start method, which significantly accelerates convergence by using the outcome of the ULA iteration as the starting point for the next step (Algorithm 2). What could be the disadvantages? Algorithm 2 tells us that for each training iteration, the last ULA samples of $p_{\theta_n}(z|x)$ must be memorized and *paring* of $x$. On the other hand, deep learning models commonly rely on mini-batch algorithms with data shuffling to handle large-scale datasets. This pairing structure conflicts with the common practice of shuffling datasets $\{x_i\}_{i=1}^{N}$, since $\{z_i^{(n)}(x)\}_{i=1}^{M}$ must be moved in tandem with the shuffling of $x$. In short, the warm-start method does not integrate efficiently with the conventional mini-batch approaches.

### 3.3 Amortized Adaptive Importance Sampling (AAIS)

Table 1: Summary of pros and cons of VAE, ALMOND, and AAIS (Proposed)

|  | **VAE** | **ALMOND** | **AAIS (Proposed)** |
| --- | --- | --- | --- |
| Efficiency | Good | Bad | Good |
| Accuracy | Bad | Good | **Superior** |

VAE and ALMOND have been reviewed so far, and their pros and cons are summarized in Table 1. We aim to leverage the advantages of each method while mitigating its disadvantages. The ALMOND objective (6) was extremely attractive, whereas all drawbacks mentioned in Section 3.2 came from the ULA sampling. Thus, the key idea is to retain the ALMOND objective while avoiding the use of the estimator (10).

### 3.3.1 Importance sampling

Importance sampling is a Monte Carlo method for estimating properties of a target distribution when direct sampling is infeasible. Instead, it uses a proposal distribution, which is a known distribution from which sampling is easier. The expected value of a function $f(z)$ under the target distribution $p(z)$ can be estimated using the importance sampling estimator as follows:

$$\mathbb{E}_{z \sim p(z)}[f(z)] = \mathbb{E}_{z \sim q(z)}\left[\frac{p(z)}{q(z)}f(z)\right] \approx \frac{1}{M}\sum_{i=1}^{M}\frac{p(z_i)}{q(z_i)}f(z_i), \tag{11}$$

where $\{z_i\}_{i=1}^{M}$ are i.i.d. samples drawn from the proposal distribution $q(z)$, and $M$ is the number of samples. This technique enables us to estimate the properties of the target distribution without its explicit form or direct access.

### 3.3.2 Proposed method

Section 3.2 underscored the difficulty of directly sampling from $p_\eta(z|x)$. We propose using importance sampling to estimate the expectation in Equation (7):

$$\mathbb{E}_{p_\eta(z|x)}[\log p_\theta(x|z)] = \mathbb{E}_{q(z)}\left[\frac{p_\eta(z|x)}{q(z)}\log p_\theta(x|z)\right] \approx \frac{1}{M}\sum_{i=1}^{M}\frac{p_\eta(z_i|x)}{q(z_i)}\log p_\theta(x|z_i), \tag{12}$$

where $\{z_i\}_{i=1}^{M}$ are i.i.d. samples drawn from a proposal distribution $q(z)$. However, two major challenges arise when using Equation (12) for estimation:

- How to calculate $p_\eta(z_i|x)$? More specifically,

$$p_\eta(z_i|x) = \frac{p_\eta(x, z_i)}{p_\eta(x)} = \frac{p_\eta(z_i)p_\eta(x|z_i)}{p_\eta(x)},$$

  which includes the intractable term $p_\eta(x)$.

- Which proposal distribution should we choose? The choice of proposal distribution is critical in importance sampling. Although Equation (11) remains an unbiased estimator regardless of choice, the convergence rate of the estimation depends on the proposal distribution (Robert (1999), Chap. 3). In other words, similar to the importance of selecting hyperparameters in the ULA-based method, choosing an appropriate proposal distribution is equally crucial.

To address the first issue, we propose replacing standard importance sampling with self-normalized importance sampling:

$$\mathbb{E}_{q(z)}\left[\frac{p_\eta(z|x)}{q(z)}\log p_\theta(x|z)\right] \approx \frac{1}{\sum_{i=1}^{M}\frac{p_\eta(z_i|x)}{q(z_i)}}\sum_{i=1}^{M}\frac{p_\eta(z_i|x)}{q(z_i)}\log p_\theta(x|z_i)$$

It can circumvent the need to compute the intractable term $p_\eta(x)$ by using the Bayes rule:

$$\frac{1}{\sum_{i=1}^{M}\frac{p_\eta(z_i|x)}{q(z_i)}}\sum_{i=1}^{M}\frac{p_\eta(z_i|x)}{q(z_i)}\log p_\theta(x|z_i) = \frac{1}{\sum_{i=1}^{M}\frac{p_\eta(z_i)p_\eta(x|z_i)}{q(z_i)}}\sum_{i=1}^{M}\frac{p_\eta(z_i)p_\eta(x|z_i)}{q(z_i)}\log p_\theta(x|z_i) \tag{13}$$

Note that this estimator is not unbiased. Nevertheless, since $\frac{1}{M}\sum_{i=1}^{M}\frac{p_\eta(z_i|x)}{q(z_i)} \to 1$ a.s. (by the Strong Law of Large Numbers), it is consistent. Furthermore, self-normalizing importance sampling is known to be

more stable than standard importance sampling and substantially contributes to handling the finite variance problem of importance sampling (Robert (1999), Chap. 3).

The second issue is more challenging. What we must not forget is the harmony between importance sampling and variational inference. In other words, we need to allow the samples from the proposed distribution $q(z)$ to serve as those from the variational distribution $p_\eta(z|x)$ at the same time. Efficiently obtaining posterior samples is essential; otherwise, we would face the same situations as in ALMOND. Thus, the desired approach will become similar to constructing a tractable variational distribution analogous to VAE. We develop the Amortized Adaptive Proposal Distribution $q_\phi(z|x)$ (AAPD), which employs a Bayesian neural network to estimate the parameters of a normal distribution:

$$q_\phi(z|x) \sim \mathcal{N}(\mu_\phi(x), \text{Diag}(\sigma_\phi^2(x)_1, ..., \sigma_\phi^2(x)_d))$$

While AAPD plays a role analogous to the encoder in a VAE, it is also trained to act as a suitable proposal distribution. Therefore, our estimator of the expectation in Equation (7) becomes

$$\mathcal{L}(\theta, \eta, \phi; x) = \mathbb{E}_{p_\eta(z|x)}[\log p_\theta(x|z)] \tag{14}$$

$$= \mathbb{E}_{q_\phi(z|x)}\left[\frac{p_\eta(z|x)}{q_\phi(z|x)} \log p_\theta(x|z)\right] \tag{15}$$

$$\approx \frac{1}{\sum_{i=1}^{M} \frac{p_\eta(z_i)p_\eta(x|z_i)}{q_\phi(z_i|x)}} \sum_{i=1}^{M} \frac{p_\eta(z_i)p_\eta(x|z_i)}{q_\phi(z_i|x)} \log p_\theta(x|z_i) \tag{16}$$

We call the proposed importance sampling method Amortized Adaptive Importance Sampling (AAIS). Importantly, the objective (14) must be optimized not only with respect to $\theta$ but also with respect to $\phi$. Yet, the objective (15) is not complete with respect to the parameter $\phi$, since it is a reconstruction loss of the autoencoder.

---

**Algorithm 3** Amortized Adaptive Importance Sampling (AAIS)

1: Initialize parameters $\theta_0$, $\phi_0$, and number of samples $M$ for the estimator in Equation (16)
2: **while** $\mathcal{L}(\theta_n, \eta_n, \phi_n; x)$ is not converged **do**
3:     $g(\theta) \leftarrow \mathcal{L}(\theta, \theta_n, \phi_n; x)$
4:     Sample $\{z_i\}_{i=1}^{M} \sim q_{\phi_n}(z \mid x)$
5:     Estimate $g(\theta)$ using the samples $\{z_i\}_{i=1}^{M}$ via Equation (16)
6:     $\theta_{n+1} \leftarrow \arg\max_\theta g(\theta)$
7:     $\phi_{n+1} \leftarrow \arg\max_\phi \mathcal{L}(\theta_{n+1}, \theta_{n+1}, \phi; x)$
8: **end while**
9: **return** optimal parameters $\theta^*, \phi^*$

---

To function $q_\phi(z|x)$ as the variational distribution, a regularization term is necessary. Adding a KL-type regularization facilitates the desired training of AAPD. Our objective is finally defined as:

$$\mathcal{L}(\theta, \eta, \phi; x) = \mathbb{E}_{q_\phi(z|x)}\left[\frac{p_\eta(z|x)}{q_\phi(z|x)} \log p_\theta(x|z)\right] - \beta D_{KL}(q_\phi(z|x)||p_\theta(z)), \tag{17}$$

where $\beta > 0$ is a weight. Since $q_\phi(z|x)$ was designed as a normal distribution, the KL divergence in Equation (17) can be efficiently computed as in Equation (5). A summarized pseudo-code is given in Algorithm 3.

Additionally, the weight $\beta$ has a role similar to that in $\beta$-VAE (Higgins et al., 2017), but it also offers another interpretation: A larger $\beta$ strengthens the KL regularization, focusing the training on the variational approximation. Conversely, a smaller $\beta$ emphasizes the reconstruction term (the expectation in (17)), directing the training toward the proposal distribution. Therefore, scheduling $\beta$ throughout training is important. One strategy for this is to employ several findings on $\beta$ scheduling in $\beta$-VAE. The adopted scheduling is described in Section 4.1.

We evaluated the performance of AAIS with an illustrative example. Consider a simple synthetic dataset, which consists of points on a 2D circle. To evaluate the effectiveness of importance sampling, we trained

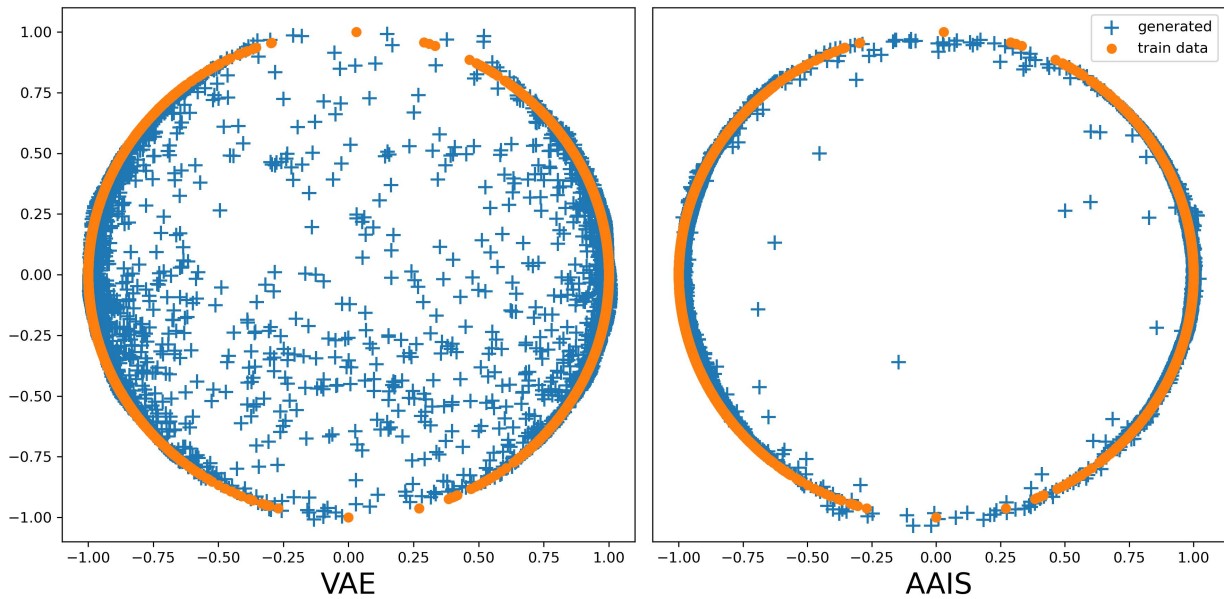

Figure 1: A comparison of data generation. (Left) VAE and (Right) AAIS.

AAIS with $\beta = 1$ (objective (17)) and compared it to VAE (objective (3)) using the same synthetic dataset. Both models shared the same model architecture, differing only in the use of importance sampling. After training, the generated points of each model are shown in Figure 1. Orange circles indicate the training data and blue pluses represent generated points. AAIS produces more precise results than VAE, demonstrating the effectiveness of importance sampling. For details, see Section 4.2.1

### 3.3.3 Practical considerations for importance weights

A common issue in this framework is that all importance weights in (16) may collapse to zero due to degenerate conditional likelihood values $p_\eta(x|z) \approx 0$. For instance, if a data point $x$ is an outlier, its corresponding estimate might not be computed during training. In this case, we switch from importance sampling to the standard Monte Carlo estimator. Specifically, if $\sum_{i=1}^M \frac{p_\eta(z_i)p_\eta(x|z_i)}{q_\phi(z_i|x)} \approx 0$ , then

$$\mathbb{E}_{q_\phi(z|x)}\left[\frac{p_\eta(z|x)}{q_\phi(z|x)}\log p_\theta(x|z)\right] \approx \frac{1}{M}\sum_{i=1}^M \log p_\theta(x|z_i), \tag{18}$$

where $\{z_i\}_{i=1}^M$ are i.i.d. samples from $q_\phi(z|x)$.

Similarly, inappropriate initialization of neural networks can cause importance weights to collapse to zero due to underflow. The remedy based on Equation (18) is ineffective because it reduces AAIS to VAE. While various methods have been developed to adjust such degenerate likelihood values (Kimura & Hino, 2024), the power weighting method (Shimodaira, 2000) is suitable for our approach: For a $\lambda \in (0, 1]$,

$$\mathbb{E}_{q_\phi(z|x)}[\frac{p_\eta(z|x)}{q_\phi(z|x)}\log p_\theta(x|z)] \approx \frac{1}{\sum_{i=1}^M (\frac{p_\eta(z_i)p_\eta(x|z_i)}{q_\phi(z_i|x)})^\lambda}\sum_{i=1}^M (\frac{p_\eta(z_i)p_\eta(x|z_i)}{q_\phi(z_i|x)})^\lambda \log p_\theta(x|z_i) \tag{19}$$

If $\lambda$ is sufficiently small, it alleviates the impact of incorrect likelihood values. During the early stages of training, the encoder $q_\phi(z|x)$ is trained more intensively than the decoder $p_\theta(x|z)$, which models the likelihood. Ideally, it is recommended for $\lambda$ to gradually approach 1. However, we observed that training was achieved even when $\lambda$ was fixed as a constant.

# 4 Experiment

This section presents a comparison of AAIS against various baselines on both synthetic and real-world datasets. The baselines include VAE (Kingma & Welling, 2014), Importance Weighted Autoencoder (IWAE) (Burda et al., 2015), Annealed Importance Sampling VAE (AMCVAE) (Thin et al., 2021), Langevin Monte Carlo VAE (LMCVAE) (Thin et al., 2021), and ALMOND (Qiu & Wang, 2021). Although ALMOND is theoretically applicable to any type of data, its implementation on datasets outside the original paper was challenging, mainly due to difficulties in tuning the ULA parameters. Therefore, for synthetic data analysis, we repeat the simulation studies described in the original paper of ALMOND.

For real-world datasets, we train AAIS and the baselines on training images and assess their reconstruction performance on the test images. Furthermore, since our model has an encoder $q_\phi(z|x)$, it can be applied to dimensionality reduction for single-cell omics datasets (which are typically high-dimensional). To compare reduction performance, we use t-distributed Stochastic Neighbor Embedding (t-SNE) (van der Maaten & Hinton, 2008), Uniform Manifold Approximation and Projection (UMAP) (McInnes et al., 2018), Pairwise Controlled Manifold Approximation and Projection (PaCMAP) (Wang et al., 2021), and single-cell Variational Inference (scVI) (Lopez et al., 2018). Finally, we explore using our model as a mixed-effect model, highlighting its potential for addressing statistical problems.

## 4.1 Experiment setup

The AAIS neural networks consist of linear layers, the Parametrized Rectified Linear Unit activation function (He et al., 2015), and layer normalization (Ba et al., 2016). Experimentally, we observed that the layer normalization effectively mitigates the issue of importance weights collapsing to zero, as mentioned in Section 3.3.3. We added skip connections (He et al., 2016) to the encoder/decoder networks for some experiments. Motivated by Parhi & Nowak (2023), weight decay was applied selectively to parts of the skip connection architecture. In all experiments, we used 100 samples for importance sampling (or Monte Carlo sampling) and Adam (Kingma & Ba, 2015) was used for optimization. The hyperparameter $\beta$ in Equation (17) was set to 1 or be scheduled according to Fu et al. (2019). A table summarizing the hyperparameters is provided in Appendix A.

For a fair comparison, all baselines, except ALMOND, were implemented with the same model architecture as AAIS (See Appendix A). The original ALMOND implementation is available[1], but the code written in MXNet (Chen et al., 2015) was too outdated. Thus, we rewrite ALMOND using PyTorch (Paszke et al., 2019)[2]. The model architecture and hyperparameters were set to the suggested ones in the original code because hyperparameter tuning for the new model architecture was prohibitively complex. The hyperparameters not mentioned for the baselines are set as default values or explained in a later section. To see training stability, we repeated each experiment 10 times for quantitative evaluation and reported the mean and standard deviation of each evaluation metric.

## 4.2 Synthetic data

### 4.2.1 Circle generation

For a generation task, we generated synthetic samples from a mixture of two distributions on a circle. Mathematically, let

$$\theta_1 \sim \mathcal{N}(0, 0.4^2) \text{ and } \theta_2 \sim \mathcal{N}(\pi, 0.4^2)$$

We sampled 2,000 points from $\theta_1$ and 3,000 points from $\theta_2$, which were then transformed into

$$(\cos\theta_1, \sin\theta_1) \text{ and } (\cos\theta_2, \sin\theta_2).$$

This transformation yields 5,000 points distributed on a circle.

---

[1]https://github.com/yixuan/almond

[2]blinded

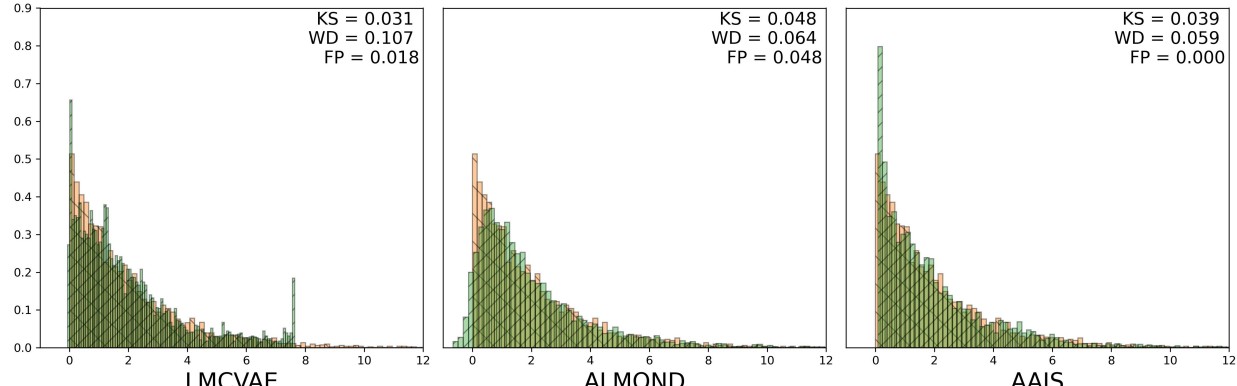

Figure 2: Signal variable visualization on $u_i \sim \exp(2)$. (Left-to-right) LMCVAE, ALMOND, and AAIS. The orange histogram represents samples $u_i$, while the green histograms display the samples generated by each trained model. Quantitative evaluation values are provided in the upper-right corner of each panel.

Table 2: Signal variable evaluation on $u_i \sim \exp(2)$. Each experiment was repeated 10 times, and the mean and standard deviation of the results were reported. The best result is highlighted in bold.

| Method | VAE | IWAE | AMCVAE | LMCVAE | ALMOND | AAIS |
|---|---|---|---|---|---|---|
| KS ($\downarrow$) | 0.092 (0.004) | 0.105 (0.013) | 0.084 (0.015) | 0.070 (0.041) | 0.048 (0.001) | **0.041** (0.011) |
| WD ($\downarrow$) | 0.198 (0.016) | 0.457 (0.094) | 0.187 (0.031) | 0.161 (0.042) | 0.068 (0.018) | **0.065** (0.006) |
| FP ($\downarrow$) | 0.092 (0.004) | 0.008 (0.004) | 0.083 (0.015) | 0.022 (0.028) | 0.048 (0.001) | **0.000** (0.000) |

We trained VAE and AAIS under the same conditions. In this setup, we assumed the conditional likelihood $p_\theta(x|z)$ follows a normal distribution with a fixed variance of 0.01 during training. The results in Figure 1 demonstrate the following: (i) qualitatively, AAIS captures the structure of the training data better than the VAE; (ii) the AAPD works effectively and facilitates smooth estimation in Equation (16); and (iii) the results emphasize the importance of accurate log-likelihood estimation.

### 4.2.2 Signal Reconstruction

Throughout Section 4.2.2, 4.2.3, and 4.3.3, we consider a scenario where the collected data is contaminated by noise. The goal is to recover the uncontaminated data distribution under the assumption that the noise structure is known. For example, let $u_i$ denote the uncontaminated variables, and let $\epsilon_i$ be i.i.d. samples from $\mathcal{N}(0,1)$. Suppose the observed data $\{x_i\}_{i=1}^n$ is defined as $x_i := u_i + \epsilon_i$. If the conditional distribution $x_i|z$ follows $\mathcal{N}(u_i(z), 1)$, methods based on variational inference can effectively denoise $x_i$. Within this framework, the decoder network $p_\theta(x_i|z)$ estimates $u_i$, which is considered a signal variable unless otherwise specified.

For evaluation, we compare the Kolmogorov-Smirnov statistic (KS) (Hodges, 1958) and the Wasserstein Distance (WD) (Ramdas et al., 2017) between the true samples $u_i$ and those generated by trained models. In particular, if the true distribution of $u_i$ is non-negative, we calculate the proportion of generated samples with negative values, referred to as the False Positive (FP) rate.

First, we sampled $u_i \sim \exp(2)$ for $i = 1, \ldots, 5000$. Next, we generated 5,000 i.i.d. samples $\epsilon_i \sim \mathcal{N}(0,1)$ and added them to obtain 5,000 contaminated samples $x_i$. Using this data, we trained AAIS alongside the baseline models. After training, we compared the true samples $u_i$ with the generated 5,000 samples from each trained model (Figure 2). The tail of the distribution learned by LMCVAE was truncated around $x = 8$, and

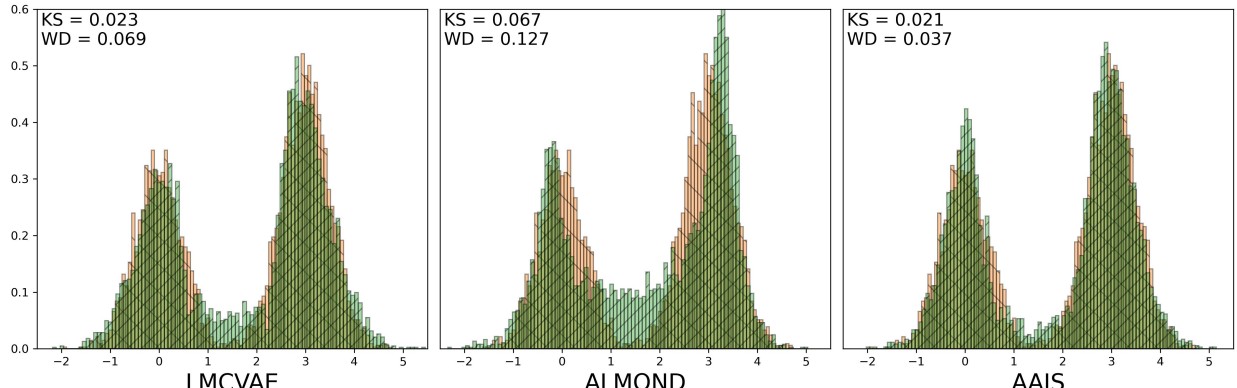

Figure 3: Signal variable visualization on $u_i \sim$ (20). (Left-to-right) LMCVAE, ALMOND, and AAIS.

Table 3: Signal variable evaluation on $u_i \sim$ (20).

|  | VAE | IWAE | AMCVAE | LMCVAE | ALMOND | AAIS |
|---|---|---|---|---|---|---|
| KS ($\downarrow$) | 0.127 (0.005) | 0.087 (0.020) | 0.123 (0.039) | **0.030** (0.007) | 0.063 (0.004) | **0.030** (0.008) |
| WD ($\downarrow$) | 0.358 (0.009) | 0.242 (0.044) | 0.346 (0.093) | 0.079 (0.008) | 0.127 (0.006) | **0.063** (0.018) |

ALMOND struggled to capture the positive domain of the exponential distribution. The green histogram generated by our model is visually the closest to the orange histogram, which represents the true samples $u_i$. Furthermore, the evaluation metrics in the upper-right corner show that our model achieved the best results. We also demonstrated the superiority of AAIS through both visual results and repeated experiments (Table 2). AAIS accomplished the most accurate estimation in terms of all metrics. Its improved performance appears to stem from reduced bias in the estimation (16). Remarkably, the AAIS perfectly learned the strictly positive domain across the 10 experiments, indicating evidence of reducing bias.

Now, we test the denoising power with a more complex form of $u_i$. Assuming

$$u_i \sim 0.4\mathcal{N}(0, 0.5^2) + 0.6\mathcal{N}(3, 0.5^2), \quad i = 1, \dots, 5000, \tag{20}$$

we repeated the same experiment (Figure 3). First, ALMOND demonstrated some learning but struggled to accurately capture the central density. This phenomenon was also observed in Figure 1 of the original paper. In contrast, LMCVAE and AAIS effectively learned the low-density area in the center and successfully estimated the two modes of the true distribution (20). Both models approximated the true distribution almost perfectly. However, a clear difference is observed in the WD metric. Quantitatively, AAIS consistently outperformed the other models (Table 3). While LMCVAE performed comparably to our model, it fell short on the WD metric. Notably, LMCVAE relies on ULA-based estimation, which significantly increases implementation time. In contrast, AAIS avoids such chain (or pairing) structures, making it faster than other ULA-based ones.

### 4.2.3 Copula modeling

This section considers a case where both $u_i$ and the noise have highly complex forms compared to Section 4.2.2. We examine a five-dimensional true signal variable $u_i$ sampled from a Clayton copula with $\theta = 2$ (generated by $\phi_2(t) = t^{-2} - 1$). Each marginal distribution follows a Gamma$(2, 1)$ distribution shifted to have a mean of zero (denoted by $G_0$). Mathematically, the distribution of $u_i = (u_{i1}, \dots, u_{i5})^T$ is

$$P(u_{i1} \leq u'_1, \dots, u_{i5} \leq u'_5) = C(F(u'_1), \dots, F(u'_5)), \tag{21}$$

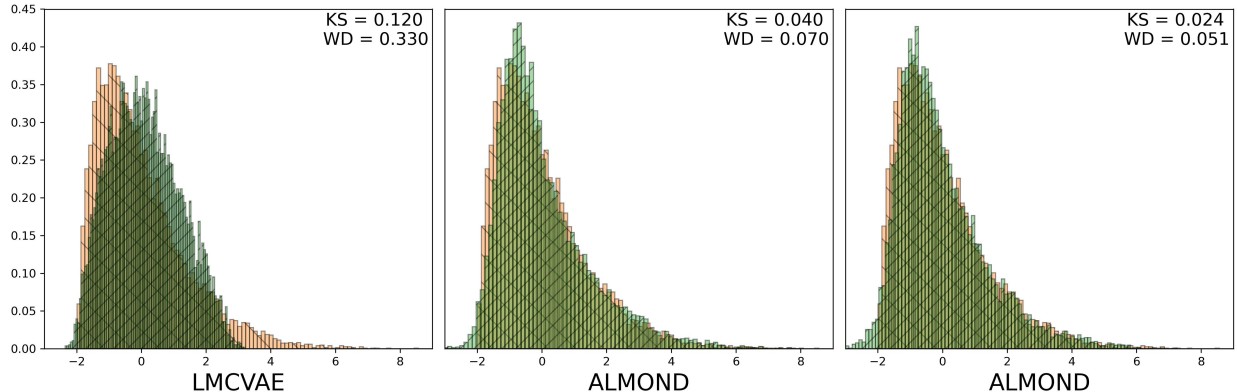

Figure 4: Signal variable visualization on $u_i \sim$ Equation (21). (Left-to-right) LMCVAE, ALMOND, and AAIS. Since $u_i$ is five-dimensional, the orange histogram represents one component of $u_i$'s. The green histograms display the same component of the samples generated by each trained model.

Table 4: Signal variable evaluation on $u_i \sim$ Equation (21). KS and WD metrics were calculated for each component of $u_i$, and their averages are reported.

|  | VAE | IWAE | AMCVAE | LMCVAE | ALMOND | AAIS |
|---|---|---|---|---|---|---|
| KS ($\downarrow$) | 0.108 | 0.089 | 0.300 | 0.151 | 0.052 | **0.043** |
|  | (0.053) | (0.028) | (0.079) | (0.339) | (0.023) | (0.017) |
| WD ($\downarrow$) | 0.313 | 0.304 | 0.677 | 0.530 | 0.153 | **0.123** |
|  | (0.105) | (0.083) | (0.125) | (0.157) | (0.070) | (0.054) |
| $L_1$ ($\downarrow$) | 0.110 | 0.043 | 0.044 | 0.056 | 0.034 | **0.033** |
|  | (0.007) | (0.008) | (0.022) | (0.028) | (0.005) | (0.007) |

where $C$ is the Clayton copula with $\theta = 2$ and $F$ is the cumulative distribution function of $G_0$. To make a more complex structure, we consider the following relationships:

$$\beta = (\beta_1, \ldots, \beta_5)^T \sim \text{Uniform}_5(-2, 2) \tag{22}$$

$$x_i, w_i \sim \mathcal{N}_5(0_5, 0.2^2 I_5) \tag{23}$$

$$\lambda_i | u_i = e^{x_i^T \beta + w_i^T u_i} \tag{24}$$

$$y_i \sim \text{Poisson}(\lambda_i | u_i) \tag{25}$$

Using (21)-(25), we sampled $\beta$ once and generated 10,000 instances of $u_i, x_i, w_i$, and $y_i$. In this setup, $\{y_i\}_{i=1}^{10000}$ is the training dataset and $(x_i, w_i)$ is considered as the covariates of $y_i$. With the addition of the covariates, we define a covariate version of the loss (17).

$$\mathcal{L}(\theta, \eta, \phi; y | x, w) = \mathbb{E}_{q_\phi(z|y,x,w)} \left[ \frac{p_\eta(z|y,x,w)}{q_\phi(z|y,x,w)} \log p_\theta(y|z,x,w) \right] - \beta D_{KL}(q_\phi(z|y,x,w) || p(z)),$$

where the decoder $p_\theta(y|z, x, w)$ estimates the true signal variables $u_i$, and then the Poisson parameter $\lambda_i$ is estimated via Equation (24).

As previously mentioned, $u_i$ follows a highly complex distribution with a complicated correlation structure, and the noise does not have the form of a specific distribution. There are two main points of evaluation. The marginal distributions of generated points from each trained model are aligned with those of $u_i$, that is, each marginal distribution should resemble a $G_0$ distribution. The alignment is quantitatively assessed using the KS and WD metrics. At the same time, $u_i$ has a nontrivial correlation structure, which is evaluated by the characteristic function of the Archimedean copula (Nelsen, 2010). The true characteristic function $\lambda(t)$ for

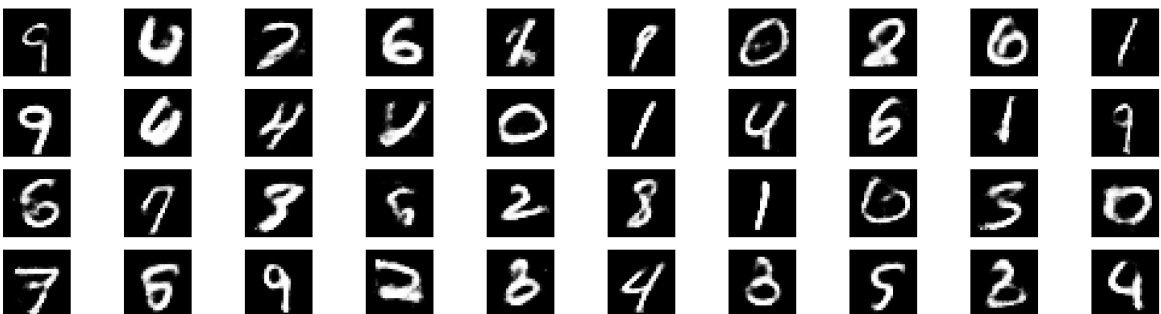

Figure 5: Generated images of AAIS trained on the MNIST training dataset

Table 5: Reconstruction error comparison on the MNIST test dataset.

|  | VAE | IWAE | AMCVAE | LMCVAE | AAIS |
|---|---|---|---|---|---|
| RE ($\downarrow$) | 62.524 (0.389) | 56.674 (0.193) | **55.598** (0.264) | 56.237 (0.176) | 56.733 (0.302) |

the Clayton copula is

$$\lambda(t) = \frac{\phi_2(t)}{d\phi_2(t)/dt} = -\frac{1}{2}(t - t^3), \quad 0 \le t \le 1$$

We will estimate the characteristic function using Theorem 4.3.4 from Nelsen (2010) (denoted by $\hat{\lambda}(t)$) and compare it to $\lambda(t)$. To quantify the difference between the two functions, $L_1$ distance was used:

$$L_1(\lambda(t), \hat{\lambda}(t)) = \int_0^1 |\lambda(t) - \hat{\lambda}(t)| dt$$

As shown in Figure 4, LMCVAE failed to adequately learn the true marginal distribution $G_0$, appearing closer to a normal distribution than a gamma distribution. In contrast, both ALMOND and AAIS successfully captured the $G_0$ distribution, while AAIS demonstrates slightly better performance on the KS and WD metrics. Table 4 demonstrates that AAIS achieves the best performance in estimating the marginal distribution. When capturing the correlation structure, ALMOND and AAIS show the nearly best performance, suggesting convergence to an optimal point. This outcome may be due to a limitation of the objective in (7).

For the generated samples $y_i$, a small probability of extremely large values can result in outliers, preventing the calculation of importance weights. To address this, we used the standard Monte Carlo method in Equation (18).

### 4.3 Real-world data

#### 4.3.1 Image generation

The MNIST dataset (Deng, 2012) consists of $28 \times 28$ grayscale images with pixel values ranging from 0 to 255. We normalized the pixel values by dividing by 255, converting them to the range $[0, 1]$, and rounding them to obtain binary values (0s and 1s). The dataset was split into 60,000 training images and 10,000 testing images for evaluation. Assuming that the generation network $p_\theta(x|z)$ follows a Bernoulli distribution, we trained AAIS, and the generated images are shown in Figure 5.

It is important to note that the AAIS objective (17) may not represent an ELBO. If $q_\phi(z|x)$ fails to adequately approximate $p_\eta(z|x)$, $D_{KL}(q_\phi(z|x)||p(z))$ will deviate from $D_{KL}(p_\eta(z|x)||p(z))$, which forms part of the original ELBO (7). As a result, directly comparing our loss function with the ELBOs of the baselines may

Table 6: Clustering performance comparison on the latent vectors of the cortex data.

|  | VAE | t-SNE | UMAP | PaCMAP | scVI | AAIS |
|---|---|---|---|---|---|---|
| ARI ($\uparrow$) | 0.584 (0.025) | 0.572 (0.034) | 0.638 (0.030) | 0.626 (0.065) | 0.699 (0.056) | **0.702** (0.033) |
| NMI ($\uparrow$) | 0.608 (0.023) | 0.611 (0.027) | 0.680 (0.022) | 0.680 (0.024) | 0.685 (0.041) | **0.699** (0.016) |

not be fair[3]. Even so, the reconstruction capability of a model such as (4) or (16) provides a comparable basis. Therefore, each trained model was evaluated on the MNIST test dataset to assess its ability to minimize Reconstruction Error (RE). Since ALMOND lacks an encoder, it is not directly applicable to evaluating any quantity on the test dataset. One potential solution is to perform ULA with an initial point; however, we could not identify suitable stationary points for the ULA samples.

As shown in Table 5, all models except VAE achieved nearly identical RE metrics. In particular, both ALMCVAE and LMCVAE used transition-based sampling to obtain posterior samples, starting from a point generated by a Bayesian neural network. Nonetheless, the differences in RE metric were small.

### 4.3.2 Dimensionality reduction

The cortex dataset (Zeisel et al., 2015), a gold standard in single-cell RNA sequencing data, comprises 3,005 samples from the mouse cerebral cortex categorized into 7 groups. Due to the high number of gene features, we first selected 1,200 genes using Stuart et al. (2019). Since the data consists of gene expression counts, we modeled the generation network $p_\theta(x|z)$ using a negative binomial distribution with a trainable inverse dispersion parameter. Inspired by Ding et al. (2018), we incorporated a softmax layer at the end of the decoder $p_\theta(x|z)$, scaling it by the total counts of $x$ to estimate the mean of the negative binomial distribution. In this experiment, the default initialization of neural networks in PyTorch often resulted in degenerate likelihoods. Therefore, we applied the power weighting method (19) with $\lambda = 0.0001$.

There are two types of baselines used for comparison. The first type consists of deep learning methods for variational inference, including VAE, scVI, and AAIS. Since the default latent dimension of scVI is 10, both VAE and AAIS use the same latent dimension. We provided the VAE with the same architecture and hyperparameters as AAIS. Meanwhile, scVI was implemented with a negative binomial likelihood, and we excluded the log-variational trick from scVI since this technique is useful only for omics data, not a central part of our model. The second type includes classical machine learning methods such as t-SNE, UMAP, and PaCMAP, which were applied using their default settings.

Figure 6 is a visualization of the learned latent variables of each trained model. Recall that the latent dimensions of VAE, scVI, and AAIS were set to 10, as previously said. To obtain a 2D visualization, UMAP was applied to the learned latent vectors of each trained model. Visually, the oligodendrocyte cluster in the VAE panel appears too close to the other three clusters. t-SNE indicates that the overall performance is satisfactory, but the clusters are too close together. This is because t-SNE primarily captures the local geometry of the data. In contrast, UMAP and PaCMAP focus more on the global structure of the data, resulting in better separation between the clusters. However, the astrocyte_ependymal and endothelial-mural clusters are still somewhat merged. Our model successfully separates these two clusters, and the overall visualization is better.

To quantitatively compare dimensionality reduction, K-means clustering (Lloyd, 1982) was performed on the learned latent vectors of the cortex data trained by each model. In this case, t-SNE, UMAP, and PaCMAP directly transformed the cortex data into a 10-dimensional space. Using the true label information, we can compare it to the labels returned by K-means clustering. To end this, the Adjusted Rand Index (ARI)

---

[3]Another reason for this is a note in Burda et al. (2015). They said there is no analogous trick to obtain a simpler KL divergence of models using multiple Monte Carlo samples, such as IWAE, AMCVAE, and LMCVAE. On the other hand, we utilized Equation (5).

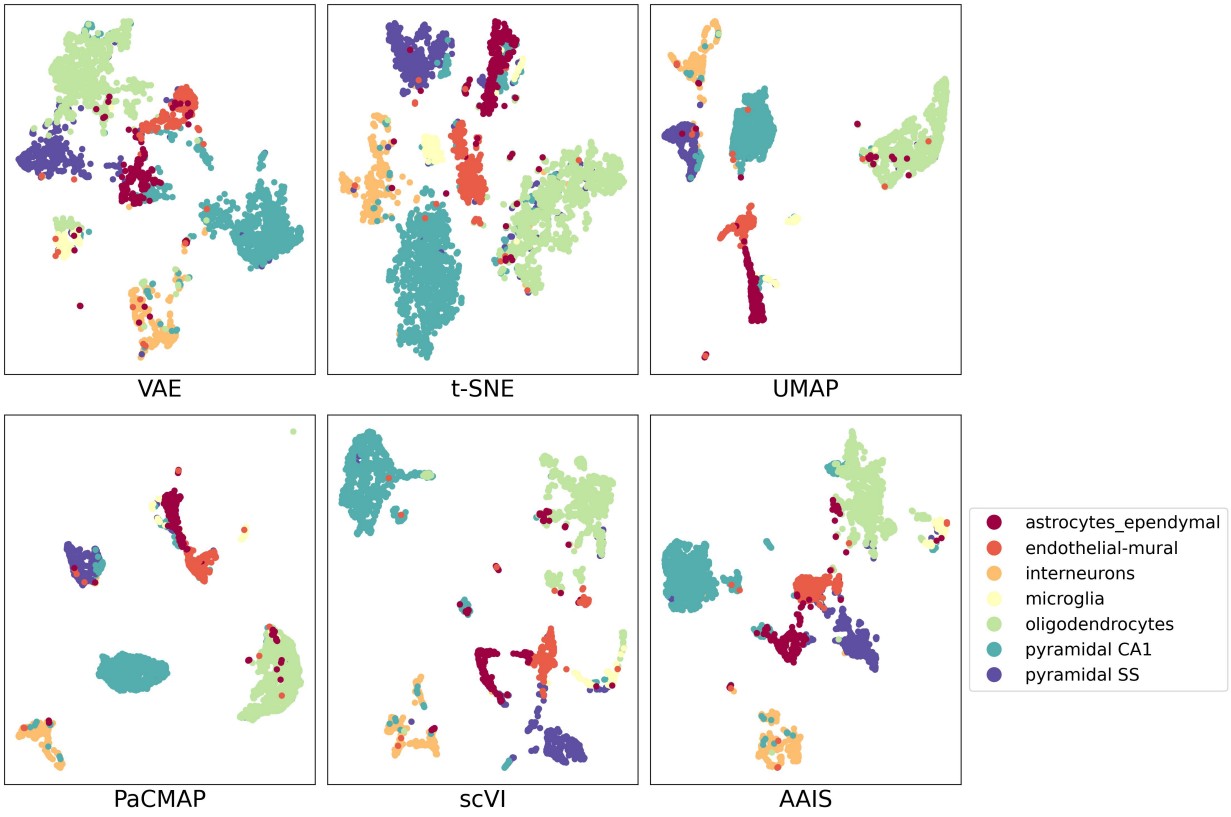

Figure 6: Cortex data visualization. (Left-top to right-bottom) VAE, t-SNE, UMAP, PaCMAP, scVI, and AAIS. The legend represents the true labels of the cortex data.

(Hubert & Arabie, 1985) and Normalized Mutual Information (NMI) (Vinh et al., 2009) scores were used for evaluation, and the results are presented in Table 6. Surprisingly, our model outperforms all the baseline models.

### 4.3.3 Application to a mix-effect model

Building on Section 4.2.3, our model can be viewed as a special case of a mixed-effects model. Recall that for each $i = 1, \ldots, K$ and $j = 1, \ldots, n_i$, a Generalized Linear-mixed Model (GLMM) is formulated by

$$g(\mathbb{E}[y_{ij}|u_i]) = x_{ij}^T \beta + w_{ij}^T u_i, \tag{26}$$

where $g$ is a link function, $y_{ij}$ is a response variable, $u_i$ is a random effect, $x_{ij}$ is a covariate, $\beta$ is a fixed effect, and $w_{ij}$ is a covariate for $u_i$. If $K$ is set to the number of data points $N$ (or if $j = 1$), there will be $N$ random effects $u_1, \ldots, u_N$ corresponding to each data point. These random effects can be used to detect unobserved stratification within data. In this case, the model (26) simplifies to:

$$g(\mathbb{E}[y_i|u_i]) = x_i^T \beta + w_i^T u_i \quad \text{for all } i, \tag{27}$$

eliminating the redundant index $j$. As an example, Equation (27) is equivalent to Equation (24) when $g$ is chosen as the log function.

A key difference, however, lies in the distributional assumption for $u_i$. In the GLMM framework, $u_i$ is typically assumed to follow a normal distribution for theoretical and practical purposes, whereas the distribution of $u_i$ was learned using deep learning techniques in Section 4.2.3, such as AAIS. This perspective allows our model to be viewed as a generalized linear mixed-effect model with non-Gaussian random effects.

Table 7: Goodness of fit comparison between GLMM and AAIS on the VerbAgg dataset. Since GLMM is not stochastically optimized, its metrics are deterministic.

|  | GLMM | AAIS |
|---|---|---|
| CNLL ($\downarrow$) | 4530.573 (NA) | **3401.038** (24.391) |
| SSPR ($\downarrow$) | 7054.688 (NA) | **5003.868** (43.739) |
| SSDR ($\downarrow$) | 9061.146 (NA) | **6802.136** (48.792) |

The VerbAgg dataset (De Boeck, 2004; Molenberghs & Verbeke, 2005), used in psychological research, is an illustrative example for exploring stratification within the data. This dataset, available through (Bates et al., 2015), contains 7,584 binary responses (*Yes* or *No*) to questions about verbal aggression. For example, one item states, *A bus fails to stop for me. I would want to curse.* The explanatory variables considered in this study are *btype*, *situ*, and *mode*.

- **btype** represents the type of behavior, with three categories: *Curse*, *Scold*, and *Shout.*

- **situ** indicates the situation, categorized as either *Other to blame* or *Self to blame.*

- **mode** differentiates between *Want* and *Do.*

We aim to determine whether the data exhibits unobserved stratification. If exists, our model will generate $u_i$ values that deviate significantly from a normal distribution. We first define a mixed-effect model as in Equation (27) with $w_i = x_i$

$$
\begin{aligned}
\text{Logit}(\mathbb{E}[y_i|u_i]) = & (\beta_{\text{intercept}} + u_{i,\text{intercept}}) \\
& + (\beta_{\text{situself}} + u_{i,\text{situself}})x_{i,\text{situself}} \\
& + (\beta_{\text{btypescold}} + u_{i,\text{btypescold}})x_{i,\text{btypescold}} \\
& + (\beta_{\text{btypeshout}} + u_{i,\text{btypeshout}})x_{i,\text{btypeshout}} \\
& + (\beta_{\text{modedo}} + u_{i,\text{modedo}})x_{i,\text{modedo}},
\end{aligned}
\tag{28}
$$

After training, we examine whether the samples generated by AAIS follow a normal distribution. As shown in Figure 7, the distribution generated by AAIS reveals two distinct modes in all panels. This suggests unobserved stratification exists within the data. One possible interpretation is that AAIS automatically estimates the variables $u_i$ without requiring explicit information about other variables. This capability stems from the potential of neural networks and the ability of AAIS to estimate a more accurate likelihood.

Why is it important to consider stratification? It enables a more accurate estimation of the model, measured by its goodness of fit. Before proceeding, it is essential to clarify that we learned the distribution of $u_i$, **NOT** a predicted value. To gain deeper insight, $u_i$ should be determined by a plausible value before further analysis. According to McCulloch et al. (2001), the conditional expectation $\mathbb{E}[u_i|y_i]$ is an optimal predictor because it minimizes the mean square error. For AAIS, importance sampling is used with the proposal distribution $u_i \sim p_{\text{AAIS}}(u)$ to obtain the random effect prediction $\hat{u}_i$:

$$
\hat{u}_i = \mathbb{E}[u_i|y_i] = \mathbb{E}_{u_i \sim p_{\text{AAIS}}(u)} \left[ \frac{p(u_i|y_i)}{p_{\text{AAIS}}(u_i)} u_i \right] \approx \frac{1}{\sum_j p(y_i|u_j)} \sum_j p(y_i|u_j)u_j,
$$

where $p_{\text{AAIS}}(u)$ stands for the distribution learned by AAIS and $p(y_i|u_j)$ is computed using Equation (28) with a Bernoulli likelihood. For GLMM, we utilized the function `ranef()` provided by the lme4 package.

After predicting the random effects $\hat{u}_i$, we evaluated the goodness of fit for each model using the Conditional Negative Log-Likelihood (CNLL), the Sum of the Squared Pearson Residuals (SSPR) (Pearson, 1900), and

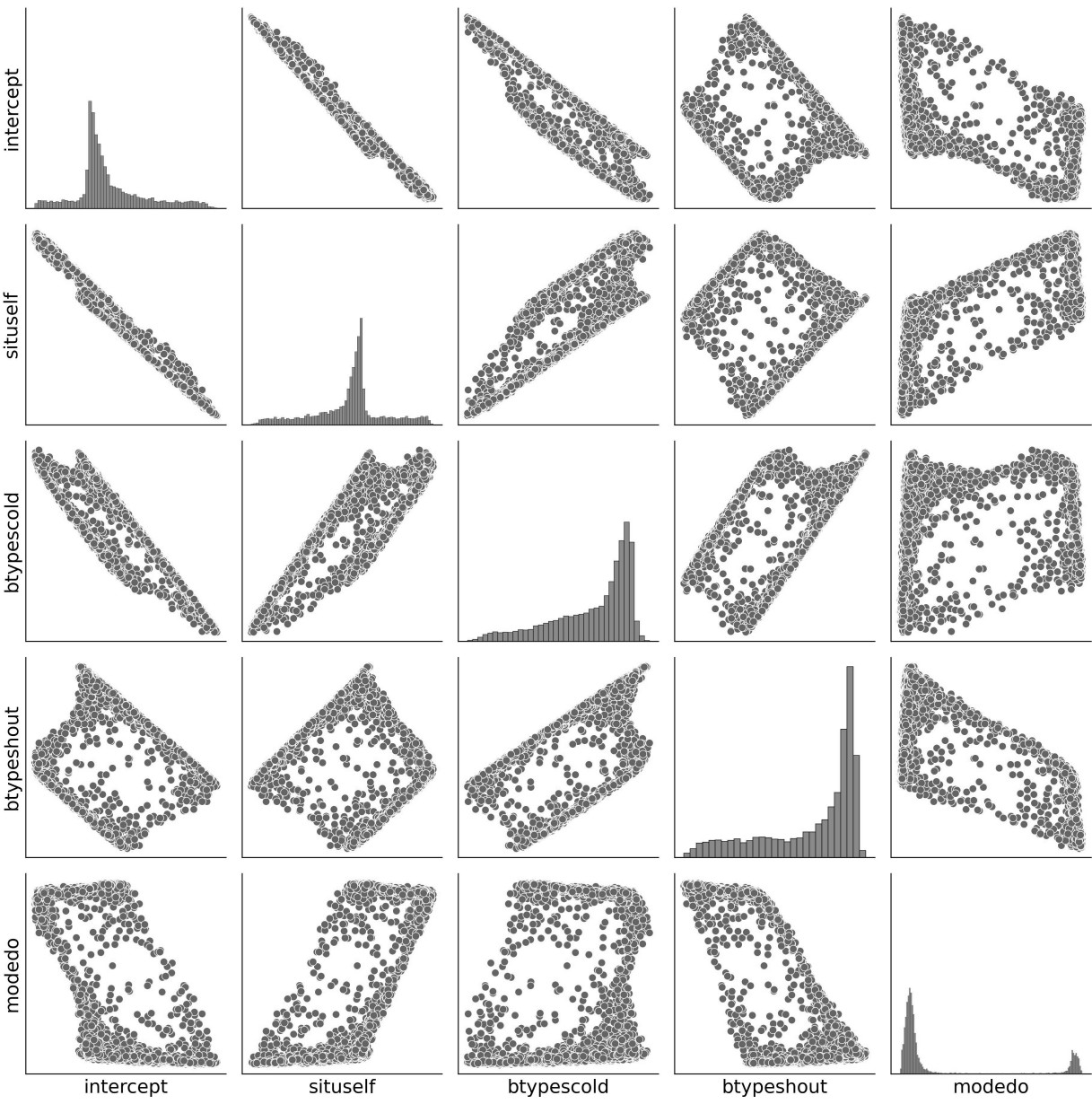

Figure 7: A pairwise plot of samples generated from AAIS trained on the VerbAgg dataset. The diagonal panels display the marginal distributions as histograms, and the off-diagonal panels show scatter plots of two components.

the Sum of the Squared Deviance Residuals (SSDR) (Nelder & Wedderburn, 1972):

$$\text{CNLL} = -\sum_i (y_i \log \hat{p}_i + (1 - y_i) \log \hat{p}_i)$$

$$\text{SSPR} = \sum_i \left( \frac{y_i - \hat{p}_i}{\sqrt{\hat{p}_i(1 - \hat{p}_i)}} \right)^2$$

$$\text{SSDR} = \sum_i \left( \text{sgn}(y_i - \hat{p}_i) \sqrt{2(\log p(y_i|y_i) - (y_i \log \hat{p}_i + (1 - y_i) \log \hat{p}_i))} \right)^2$$

In these equations, $\hat{p}_i := \mathbb{E}[y_i|\hat{u}_i]$ is derived from Equation (28), and $p(y_i|y_i)$ denotes the saturated log-likelihood. Table 7 demonstrates that AAIS outperforms GLMM, highlighting the importance of considering data stratification. Additional results and discussions can be found in Appendix B.

## 5 Conclusion

In this study, we introduced Amortized Adaptive Importance Sampling (AAIS), a novel generative model designed to enhance estimation accuracy and computational efficiency in variational inference. AAIS utilizes an importance-sampling-based framework that builds on the strengths of existing methods, such as VAE and ALMOND, while overcoming their limitations. This methodology provides small biased estimations and circumvents the drawbacks of ULA-based techniques.

Extensive experiments on synthetic and real-world datasets show that AAIS consistently outperforms traditional baselines across various metrics. In synthetic datasets, it demonstrated its effectiveness in capturing complex latent structures, including non-Gaussian and multimodal distributions. In applications, AAIS achieved comparative performance in tasks such as image generation, dimensionality reduction, and mixed-effects modeling. In particular, its extension to mixed-effects models highlights the importance of uncovering intrinsic data stratification.

There are several promising directions for future research. First, the requirement for multiple Monte Carlo samples per data point complicates the direct application of CNN modules. While we performed computations in the $(N, M, d)$ format for efficient data storage and processing (Section 4.3.1), the CNN module enforces data to be reshaped into $(NM, d)$ for compatibility. Since CNN-based neural networks are highly effective in processing image data, further research is worthwhile. Second, while the power weighting method was introduced to resolve degenerate likelihoods (Section 4.3.2), its scheduling was not explored in this study. Investigating whether scheduling can improve dimensionality reduction presents a valuable opportunity for future study. Alternatively, exploring another method could provide additional insights. Finally, we only considered a special case of GLMM. Our current model lacks a natural extension for handling replications in one cluster. Future research will aim to adapt the generative model to more general GLMM scenarios.

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

# A  Model architecture and hyperparameters

Table 8: Hyperparameter configurations used across datasets. NB stands for Negative Binomial distribution.

| Dataset | circle | exp | mix | copula | MNIST | cortex | VerbAgg |
|---|---|---|---|---|---|---|---|
| Batch size | 128 | 128 | 128 | 1000 | 128 | 128 | 1000 |
| Latent dim | 2 | 1 | 1 | 5 | 10 | 10 | 5 |
| Skip conn. (encoder) | No | No | Yes | No | No | No | Yes |
| Skip conn. (decoder) | No | No | Yes | Yes | No | No | Yes |
| # layers | 4 | 4 | 4 | 2 | 3 | 3 | 4 |
| # params (K) | 2.17 | 2.06 | 6.30 | 4.96 | 1078.05 | 688.50 | 38.94 |
| $\beta$ scheduler | No | No | No | No | Yes | Yes | No |
| Likelihood $p_\theta(x|z)$ | Normal | Normal | Normal | Poisson | Bernoulli | NB | Bernoulli |
| Learning rate | 1e-3 | 1e-4 | 1e-3 | 1e-4 | 1e-3 | 1e-3 | 1e-4 |
| Weight decay | 0 | 0 | 0.4 | 0 | 0 | 0 | 0 |
| Epochs | 100 | 1000 | 1000 | 2000 | 100 | 200 | 10000 |

Table 8 provides a summary of the hyperparameter settings for each experiment. The $\beta$ scheduler indicates whether the method proposed by Fu et al. (2019) was applied.

# B  Additional results on the VerbAgg dataset

Under the normality assumption in GLMM, the distribution $p_{\mathrm{GLMM}}(u)$ trained by GLMM cannot take a form other than a normal distribution. Figure 8 shows a pairwise plot of samples generated by GLMM. As expected, the histograms and scatterplots display typical characteristics of a normal distribution. Compared to Figure 7, GLMM fails to account for data stratification, leading to a reduction in the model's explanatory power (Cf. Table 7).

Table 9: Fixed effect estimates on the VerbAgg dataset.

| | GLMM | AAIS |
|---|---|---|
| $\beta_{\mathrm{intercept}}$ | 1.408 (0.060) | 1.593 (0.032) |
| $\beta_{\mathrm{situself}}$ | -0.816 (0.050) | -0.895 (0.011) |
| $\beta_{\mathrm{btypescold}}$ | -0.859 (0.060) | -1.085 (0.025) |
| $\beta_{\mathrm{btypeshout}}$ | -1.618 (0.063) | -2.417 (0.052) |
| $\beta_{\mathrm{modedo}}$ | -0.538 (0.050) | -0.399 (0.013) |

Table 9 summarizes the fixed effect estimates from GLMM and AAIS. Especially, the values of $\beta_{\mathrm{btypeshout}}$ differ significantly between the two models, which can be attributed to the misspecification of the random effect distribution. Previous studies on random effect misspecification (e.g., McCulloch & Neuhaus (2011a) and McCulloch & Neuhaus (2011b)) have reported that such misspecification typically does not significantly affect fixed effect estimation. However, they also noted that as the learned distribution deviates further from normality, these differences become more pronounced. Our findings regarding fixed effects are consistent with theirs, as we observed data stratification and its impact on goodness of fit (Figure 7 and Table 7).

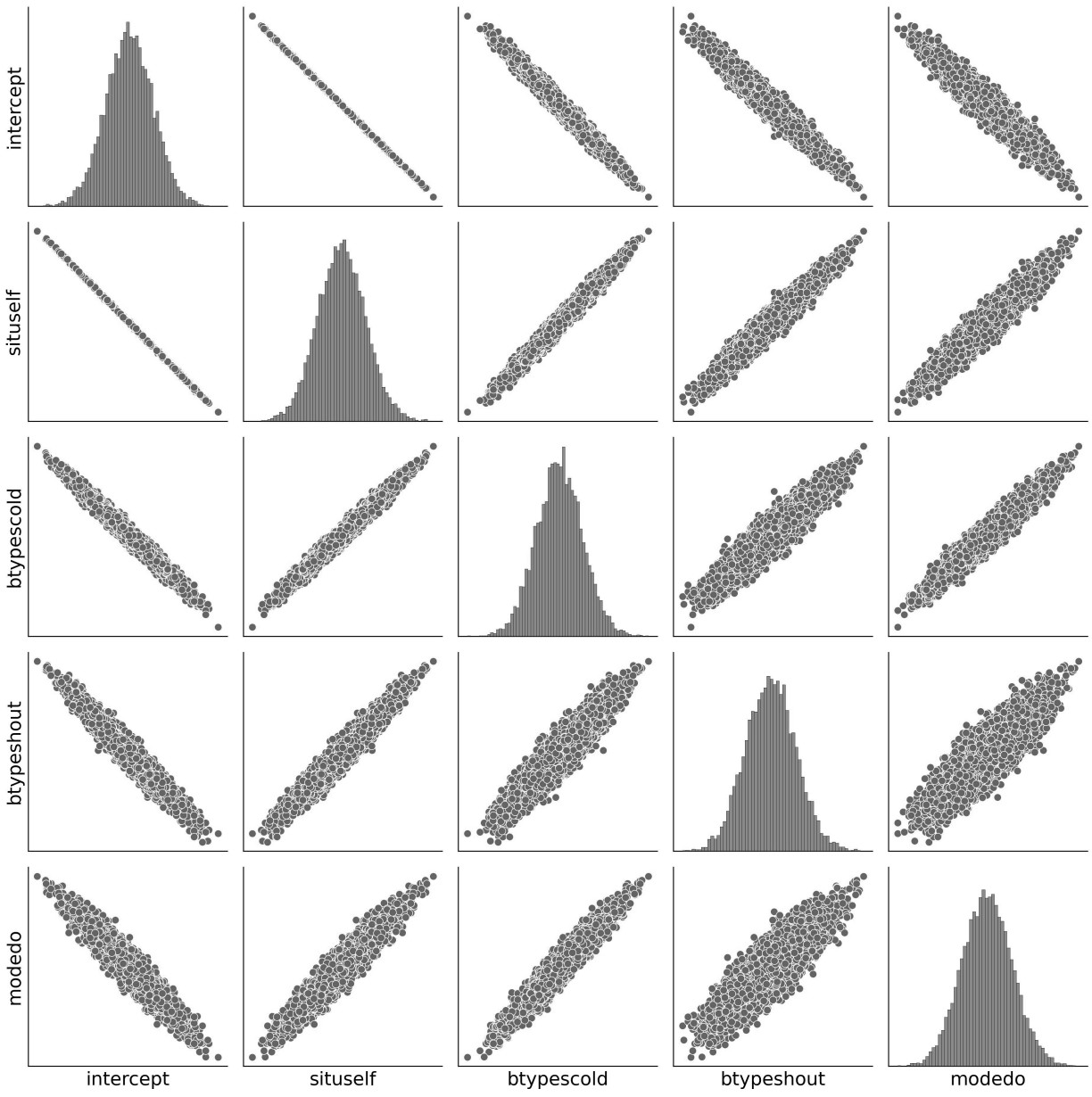

Figure 8: A pairwise plot of samples generated from GLMM trained on the VerbAgg dataset.

