# OpenReview forum: "Efficient and Accurate Likelihood Estimation via Learning Amortized Adaptive Proposal Distributions"
_TMLR — Withdrawn by Authors_

### Review · Reviewer_2YB9 · 2025-10-09

**Summary Of Contributions:**

The paper proposes a latent variable generative modeling framework that aims to improve ALMOND's computational efficiency by replacing its unadjusted Langevin dynamics (ULA) based estimator with self-normalized importance sampling (SNIS). The importance sampling proposal is learned through another neural network in an amortized fashion (similar to VAE), i.e., by predicting the mean and diagonal covariance per input $x_i$. This amortized proposal distribution is also used similarly to the variational distribution in VAE. The paper also proposes two practical heuristics for improving importance weighting during training. Experiments across synthetic tasks and real-world datasets show that the proposed approach often outperforms alternative baselines.

**Key strengths**:
- The paper is clearly motivated: it addresses the sensitivity and inefficiency of ALMOND’s ULA-based estimator and proposes an amortized SNIS alternative that is potentially more efficient and simpler to implement.
- Broad experimental coverage with generally positive results (though many experiments appear inherited directly from ALMOND).

**Key weakness**:
-  The title's claim "Accurate likelihood estimation" is not supported by likelihood estimation accuracy in the experiments and is ambiguous. How the likelihood is estimated and whether the optimization objective (Eq. 17) is a principled likelihood estimator or a bound is not clear.

**Additional Comments:**

In Algorithm 3, is $g(\theta)$ formed from Eq. 14 or Eq. 17? I assume it should be the latter, although $\max g(\theta)$ is equivalent for both equations if $p_\theta(z)$ does not depend on $\theta$ (e.g., being a standard Gaussian).

> Recent advancements in probabilistic modeling have driven significant progress in deep learning, particularly through the development of generative models based on variational inference.

This reads somewhat awkward, and recent powerful generative models are not primarily built on top of variational inference (LLM, diffusions, etc).

> Additionally, Kingma & Welling (2014) proposed the reparameterization trick to improve backpropagation
efficiency in the encoder. However, this approach is primarily applicable to Gaussian distributions.

What does it mean: "backpropagation efficiency in the encode"?
For the second part, I think it's not true: there is a range of choices other than Gaussian.

> While
VAE enables efficient posterior sampling through the reparameterization trick

It reads a bit awkward. There's no direct relation between efficient posterior sampling with the reparameterization trick.

>Qiu & Wang (2021) further proposed a novel strategy for tightening the ELBO by
parameterizing the variational distribution as an alternative representation of the posterior distribution.

This reads confusing. What does it mean exactly? In Qiu & Wang (2021), isn't the variational distribution approximated with samples with Langevin dynamics?


> We develop the Amortized Adaptive Proposal Distribution qφ(z|x) (AAPD), which employs a Bayesian neural network to
estimate the parameters of a normal distribution

It is not a "Bayesian neural network".

> This outcome may be due to a limitation of the objective in
(7).

What does this mean?

> Our current model lacks a natural extension for handling replications
in one cluster. Future research will aim to adapt the generative model to more general GLMM scenarios.

This is not very clear to readers not familiar with GLMM.

**Audience:**

Yes

**Audience Explanation:**

Yes. Readers working on VAEs and latent variable modeling would be interested in the proposed method in the paper and its empirical performance.

**Claims And Evidence:**

Yes

**Claims Explanation:**

The claims made in the submission are **partly** supported.

Empirical results indicate the method can work well in practice, but I found the main claims obscure after reading the paper (see weakness above and points below in requested changes).

> Furthermore, the importance sampling estimator is
unbiased (or has small bias), leading to more accurate estimates compared to ULA

Throughout, self-normalizing importance sampling is used, and therefore, "unbiased" is misleading. Also, I am not sure why it is more accurate than ULA, given that the importance sampling can also fail drastically in case of a poor proposal (i.e., high variance).

**Requested Changes:**

- Clarify the objective in Eq. 17. Justify why optimizing it approximates maximum likelihood estimation. Is it a lower bound to the likelihood? If no theoretical guarantee and experimental evidence exist, the claims in the title (“efficient and accurate likelihood estimation”) should be changed or moderated.
- There should be better explanations in the main text from Eq. 14 to Eq. 17: it got me confused why the KL term in the ELBO is dropped out in Eq. 14 and suddenly comes back in Eq. 17, where $p_\eta(z|x)$ is partially replaced by $q_\phi(z |x)$, with an additional hyperparameter $\beta$ introduced.
	- I think $\mathcal{L}(\theta, \dots)$ should be removed from Eq. 14 since it's only the first term in the original ELBO. And it seems only the first term is estimated with importance sampling while the second term KL divergence is approximated directly by replacing $p_\eta(z|x)$ with $q_\phi(z |x)$ without involving importance sampling.
	- If my understanding above is correct, it is not clear to me why $q_\phi(z |x)$ should also serve as the variational distribution in place of $p_\eta(z|x)$, apart from being an importance proposal for computing expectations. From Eq. 8 in Qiu & Wang (2021), $q_\phi(z |x)$ can be directly used to estimate the gradient with importance sampling, which should be all that is needed?
- Further clarify the use of stabilization heuristics (Eqs. 18–19).  Indicate whether both are used, under what conditions, and how often fallback is triggered. Report the chosen $\lambda$ values (or in the appendix, clarify for each experiment if not used) and justify deviations from Eq. 17:
	-  In my understanding, in order for the optimization objective to be consistent with Eq. 17, these two tricks should only serve as a warm-up strategy at the beginning of the optimization.
- Be consistent: Amortized Adaptive Proposal Distributions in the title, while Adaptive Amortized Proposal Distribution in the main text.
- Update references to their published venues (e.g., IWAE → ICLR 2016)
- $\beta$ in Eq. 22 is in conflict with $\beta$ in Eq. 17.
- Reconstruct error is not defined. KS, WD, FP appear in Figure 1 before their definitions.
- Eq.(1), $\log p(x)$ should be $\log p_\theta(x)$

Recommended:
- Improve literature review: It may be better to make a more thorough literature review on VAEs and latent variable modeling.

---

### Review · Reviewer_CAfz · 2025-10-21

**Summary Of Contributions:**

The paper proposed a methodology for learning static latent variable models via VAE/variational EM type approaches. An emphasis in the abstract and introduction is put on mixed-effects models. The motivation for the methodology remains a little generic, with the abstract mentioning the existing methods "often encounter trade-offs between approximation accuracy and computational efficiency".
The method proposed here is framed as an alternative/improvement to the "ALMOND" method of Qiu & Wang (2021).


- Qiu, Y. and Wang, X., 2021. Almond: Adaptive latent modeling and optimization via neural networks and langevin diffusion. Journal of the American Statistical Association, 116(535), pp.1224-1236.

**Additional Comments:**

- In 3.3.3. you mention: "In this case, we switch from importance sampling to the standard Monte Carlo estimator" - this is not very "principled" for a lack of a better term. Firstly, you should make sure your implementations are in log space (and, even for expectations of possibly negative functions such as in (18), you can still do logspace computations as libraries such as scipy implement logsumexp with a sign return option). Secondly, if the issue is not numerical but that the proposal is bad (hence weights are badly behaved), you should rather seek to improve the proposal. Weight tempering is an ok strategy, but indeed it introduces bias. The claim: "The remedy based on Equation (18) is ineffective because it reduces AAIS to VAE" -  is strange; as vanilla VAE needs no introduction or training of $ p_{\eta}(z | x) $.
- What is the "mean ergodic estimator" mentioned several times? Please explain better. Is it just MCMC?
- Please rephrase your second contribution, as it makes it look like this is the only work integrating VI with importance sampling in this context.
- In "review of literature", it would be appropriate to mention the literature on adaptive importance sampling, particularly regarding the self-normalized IS estimator that is used here.

**Audience:**

Yes

**Audience Explanation:**

I believe that the method, being a SNIS extension of ALMOND, can indeed be potentially of interest to some members of the TMLR community interested in general latent variable modelling. However, this is conditional on addressing some concerns I express in the other boxes.

**Broader Impact Concerns:**

No broader impact concerns.

**Claims And Evidence:**

No

**Claims Explanation:**

The paper lacks in terms of accurate and convincing evidence, due to vague claims and unclear important parts of the proposed method.
In fact, to my interpretation one detail of the proposed method is incorrect.
Further, discussion of important literature is missing.

- Table 1 definitely lacks in specificity. What is "efficiency" and "accuracy" here?  The main disadvantage of ALMOND that seems important is the difficulty of doing minibatching with it. However this point is not explained clearly and too quickly, it is not spelled out fully. "This pairing structure conflicts with the common practice of shuffling dataset [...] must be moved in tandem with the shuffling" - please spell out this part more. What is the core problem here? Would ALMOND give non-valid estimates? Make a concrete example
- Algorithm 3 does not refer to the $\eta$ parameters at all. I am not sure to understand what is done here. I can guess, but it is definitely not spelled out well.
- **[Potential mistake in the method]**: Eq (7) is indeed a correct re-writing of Eq (3) with the particular choice of variational distribution (VD) family given by $ p_{\eta}(z | x) $, which is exactly the same as $ p_{\theta}(z | x) $ but the use of a different variable $\eta$ denotes that this same class of densities is now used as variational distribution. Which is basically an energy-based model, as it is a variational distribution with an intractable normalizing constant, and that we cannot sample easily (unless with ULA or other MCMC schemes). To my interpretation, the proposed method says: we want to use this VD, but sampling from it with ULA is inconvenient. Therefore, we go to expectations over $ p_{\eta}(z | x) $ (like those in Eq (3) ), and apply (self-normalized) IS instead to estimate them, with another proposal $q$. This makes sense, but **[from Eq 17, it seems that the authors apply SNIS correctly to the first term of Eq (3), but in the second term they simply replace $ p_{\eta}(z | x) $ with the newly introduced $q$, instead of applying SNIS ]** - which would be the principled choice, in my own understanding of the proposed method. Also, it should be stated that the interpretation of the introduced $q$ is different from the $q$ in VAE: we want the variational distribution $p_{\eta}(z | x) $ to match $p_{\theta}(z |x)$ (as in ALMOND), while $q_{\phi}$ is simply an additional auxiliary distribution introduced due to the inconvenience that we cannot easily estimate expectations in Eq (3) otherwise. Therefore, I do not think that $q_{\phi}$ should necessarily itself match $p_{\theta}(z |x)$. Rather, the introduced $q$ should ideally lead to low-variance gradients of the objective.

**Requested Changes:**

The most critical change to address is about what I interpret as a mistake, as explained above.
Further:
- A better description of Algorithm 3, in the sense that, the interactions between the three parameters being optimized $\eta, \theta, \phi$ should be clarified.
- The first line of Eq (1) contains an incorrect equality : $\log p_\theta(x)=\mathbb{E}_{q_\phi(z \mid x)}[\log p(x)]$. Please revisit your derivation.
- Discussion of relevant related literature, where $q$ explicitly takes the role of an importance sampling proposal. There are several works on this line. See below.


References


- Kim, D., Hwang, J. and Kim, Y., 2020, June. On casting importance weighted autoencoder to an EM algorithm to learn deep generative models. In International Conference on Artificial Intelligence and Statistics (pp. 2153-2163). PMLR.
- Rainforth, T., Kosiorek, A., Le, T.A., Maddison, C., Igl, M., Wood, F. and Teh, Y.W., 2018, July. Tighter variational bounds are not necessarily better. In International Conference on Machine Learning (pp. 4277-4285). PMLR.
- Cremer, C., Li, X. and Duvenaud, D., 2018, July. Inference suboptimality in variational autoencoders. In International conference on machine learning (pp. 1078-1086). PMLR.

---

### Review · Reviewer_Sf5J · 2025-10-29

**Summary Of Contributions:**

The paper proposes Amortized Adaptive Importance Sampling (AAIS) as a latent-variable generative modeling method intended to improve posterior approximation in variational inference. The stated motivation is to retain the tighter training objective structure of ALMOND, which in their interpretation replaces the VAE’s fixed variational family by a learned posterior, while avoiding ALMOND’s reliance on Unadjusted Langevin Algorithm (ULA) sampling, which can be computationally demanding and sensitive to tuning. AAIS instead introduces a learned proposal distribution to enable self-normalized importance sampling of latent variables within an encoder–decoder architecture. The method is evaluated on several synthetic and real-world datasets, demonstrating competitive performance against VAE variants and suggesting applicability to tasks such as dimensionality reduction and mixed-effects modeling. While the paper aims to combine more accurate latent inference with improved practical efficiency over ULA-based approaches, the sparse coverage of the literature, the insufficient theoretical justification of the training objective and the empirical evidence provided do not yet fully establish the claimed likelihood improvements or computational benefits.

**Audience:**

Yes

**Audience Explanation:**

The paper addresses the important question of improving inference and generative performance in latent variable models by incorporating importance sampling into amortized variational inference. This is a topic of interest to researchers working on probabilistic modeling, variational autoencoders, and approximate inference. The applied examples suggest potential utility in areas such as single-cell RNA sequencing data analysis and mixed-effects modeling.

**Broader Impact Concerns:**

No concern.

**Claims And Evidence:**

No

**Claims Explanation:**

On the introductory sections:

* Sections 1 and 2. The Introduction and Literature Review provide broad context on deep learning and generative models, but spend significant space on standard background material.  The coverage of relevant literature is sparse, without clearly articulating the specific research gap that AAIS aims to fill within the existing body of work on variational inference and importance-weighted training objectives.
* Section 3.2. The description of ALMOND appears, in my understanding, imprecise and diverging from the cited original method. ALMOND is not a VAE with a learnable encoder; its ELBO-like structure comes from a minorize-maximize scheme using the model posterior $ p_{\tilde{\beta}}(z \mid x) $ from the previous iterate, not a parameterized variational distribution $q_{\phi}(z \mid x)$. Thus, it is unclear why ALMOND should be viewed as a variational-inference method. Additionally, ALMOND fixes the latent base distribution $\pi_0(z) $and relies on the flexibility of $ h_{\eta}(z) $; treating $\pi_0 $ as learnable or omitting updates to $ \eta $would require justification that the theoretical guarantees are preserved. Finally, since Qiu \& Wang explicitly support mini-batch stochastic gradients, the claim that warm-starting conflicts with mini-batch training needs stronger justification.

On the methodology (Section 3.3):
* Again substantial portion of Section 3.3 revisits established ideas such as importance sampling and self-normalized estimators, but does not clearly outline existing importance-weighted variational inference models. This makes it difficult to understand what is distinct in the proposed formulation with respect to known  methods.
* The final objective of Equation (17) appears to combine a self-normalized importance-weighted reconstruction term with a $\beta$-weighted KL regularization. While this seems to differ from commonly used objectives, it appears to function as a heuristic surrogate: there is no analysis establishing that it is a valid lower bound on $\log p_\theta(x)$, nor any thorough discussion of the bias and variance introduced by the self-normalized importance sampling estimator. As a result, it is unclear what quantity is actually being optimized or under what conditions the procedure can be expected to succeed. In this regard, qualitative claims such as “superior accuracy” in Table 1 are difficult to interpret.
* Although ALMOND is discussed extensively as motivation, the relationship between ALMOND’s minorize–maximize structure and the proposed objective is not clearly justified, and it is unclear whether ALMOND’s theoretical guarantees (for example, tightness at convergence) are preserved.


On the experiments (Section 4):

* The experimental section is broad, covers several synthetic and applied datasets, and includes repeated-run reporting,
which is appreciated. However, the results still do not clearly support the
core performance claims.
* While the experiments cover interesting cases, they do not evaluate density-modeling performance on established generative-modeling settings (for example, static binarized MNIST with test NLL, Omniglot/Fashion-MNIST, or the UCI tabular density suite). Including one or more of these would help readers compare.
* The choice of baselines is not clearly justified in terms of which methods are most comparable to the proposed approach or why they are the right points of reference for the tasks considered. In addition, parity of architecture, hyperparameters, and compute budgets across models is not fully documented.
* Across experiments, the reported metrics focus on latent-distribution matching (KS/WD), reconstruction error, or clustering quality (ARI/NMI), none of which seems to directly evaluate the paper’s primary claim of improved likelihood estimation or posterior accuracy on held-out data. Without a shared, likelihood-based evaluation protocol across tasks, it remains difficult to assess the extent to which the proposed method achieves its stated probabilistic objectives.
* Efficiency claims are not supported by wall-clock runtime, hardware information, time-to-target accuracy, or scaling behavior (e.g., with respect to the number of importance samples).
* The core estimator relies on self-normalized importance sampling, yet there is no analysis or diagnostic (e.g., ESS, log-weight variance) to verify weight stability or to justify that the estimator is reliable for the chosen number of samples.
* Ablations are not provided for key hyperparameters such as the number of importance samples, the proposal distribution capacity, or the $\beta$-schedule, which limits insight into robustness.

**Requested Changes:**

Major (issues that must be addressed):

1. Clarify the objective being optimized. Specify whether the proposed objective is intended to be a valid lower bound on $\log p_\theta(x)$, and if not, describe what quantity is being optimized. An analysis of the bias and variance introduced by self-normalized importance sampling would help establish when the method can be expected to perform well.
2. Refine performance claims. Terms such as “superior accuracy” should be precisely defined and linked to well-motivated evaluation criteria.

3. Clarify the relationship to ALMOND. Either demonstrate theoretically that key ALMOND guarantees (for example, tightness at convergence) are retained under the proposed modifications, or update the framing to avoid implying that such guarantees carry over automatically.

4. Improve positioning relative to prior work. Strengthen the discussion of how this approach differs from, and potentially improves upon, existing importance-weighted variational inference methods. This would help readers understand the intended contribution.
5. Align claims with evaluation metrics by reporting a shared, likelihood-based evaluation on held-out data and, where posterior inference is central, a metric that evaluates posterior accuracy or predictive performance. This can be complemented by domain-specific metrics where appropriate.
6. Provide concrete evidence supporting efficiency claims, such as wall-clock runtime and scalability comparisons (e.g.\ with respect to the number of importance samples $M$) under matched compute and hardware conditions.
7. Justify baseline comparisons, and clearly document architecture and compute-budget parity. If ALMOND cannot be tuned fairly, clarify that limitation and temper comparative claims accordingly.
8. Demonstrate the reliability and robustness of the self-normalized importance sampling component through minimal diagnostics (e.g. ESS or log-weight variance) and ablations.


Minor (improvements that would strengthen the paper):
* Improve clarity of Section 3.1. The paragraph on VAEs is long but still lacks clarity: it includes unnecessary derivations (e.g.,
Eq. (5)), tautological steps (the first two equalities in Eq. (1)) and vague statements, and does not justify why a latent-variable model helps (e.g., reduced dimension). Gaussian mean-field choices are modeling decisions, so implying VAEs fundamentally cannot maximize log-likelihood is misleading.
* Include additional standard benchmark datasets to improve comparability with prior work.

---

### Note · Authors · 2025-11-04

I have read and agree with the venue's withdrawal policy on behalf of myself and my co-authors.